# Coordinated Control of RSC and GSC for DFIG System under Harmonically Distorted Grid Considering Inter-Harmonics

**Bo Pang [1]** , **Hui Dai [2], Feng Li [2] and Heng Nian [1],***

1   College of Electrical Engineering, Zhejiang University, Hangzhou 310027, China; 11610022@zju.edu.cn
2   State Grid Huaian Power Supply Company, Huaian 223002, China; daihui.ha@js.sgcc.com.cn (H.D.);
    lifeng.ha@js.sgcc.com.cn (F.L.)
*   Correspondence: nianheng@zju.edu.cn

**Abstract:** For improving the performance of a doubly fed induction generator (DFIG) system under a harmonically distorted grid, this paper proposes a coordinated control strategy which is effective for grid inter-harmonics as well as grid integer harmonics. In order to suppress the negative impacts caused by grid harmonics, including inter-harmonics, this paper introduces an additional harmonics suppression controller, which contains a Chebyshev high-pass filter and a modified lead element considering the delay compensation. The proposed controller is employed in the rotor side converter (RSC) and grid side converter (GSC). Based on the proposed harmonics suppression controller, a coordinated control strategy between RSC and GSC is developed, in which the control targets, including the sinusoidal output current, constant power, or steady generator torque, can be achieved for DFIG, while GSC is responsible for maintaining the sinusoidal total current to guarantee the power quality of the grid connection. The effectiveness of the proposed method is verified by the theoretical analysis, and the experimental results derived using a 1 kW DFIG system validate the correctness of the theoretical analysis.

**Keywords:** coordinated control; doubly fed induction generator; inter-harmonics; harmonics suppression

## 1. Introduction

Due to the usage of power electronics devices and non-linear loads [1–4], harmonics is a subsistent problem in power grid. Institute of Electrical and Electronic Engineers (IEEE) recommends that the distortions in grid voltage can be 8% at most [2], while total harmonic distortions in grid voltage are allowed to be 5% at most in China [3]. Under the harmonically distorted grid, the doubly fed induction generator (DFIG) will output distorted current components, which can severely pollute the power grid. The grid codes indicate that the distortions in the output current have to be limited for DFIG system with the connection to grid [2,3], in which the harmonic components between the third and 11th output currents should be less than 4%, and the total harmonics distortions (THD) of current cannot beyond 5%.

For the operation of DFIG system under 5th and 7th orders grid harmonics, the traditional method based on phase sequence decomposition has the disadvantages of the inevitable control delay and complicated calculation. Therefore, to overcome this problem, multi-PI [5], PR or PIR [6], and direct resonant control (DRC) [7] have been developed to replace the traditional method. For suppressing the higher 6n ± 1 order harmonic current, a controller based on repetitive control (RC) [8–10] was developed. Furthermore, the nonlinear control methods include model predictive control and sliding

mode control are also used to suppress integer harmonics, while the performance is affected by complicated computation and parametric dependence [11].

Besides the integer harmonics, grid voltage usually contains inter-harmonic voltage components caused by adjust variable device (ASD) [4], high frequency resonance (HFR) [12,13], and so on. According to the stipulation of IEEE, the individual inter-harmonics in grid voltage can be allowed no higher than 1%–5% in different systems with different voltage levels [14–18]. Since the existing harmonic current suppression methods based on the resonant controller or RC cannot be suitable for inter-harmonics suppression due to the characteristic of frequency dependence, [18] proposed a wideband controller to suppress inter-harmonics in stator current for DFIG. However, besides the current distortions, the power and torque ripples still exist in DFIG even when the stator current is suppressed, since the torque ripples and current distortions cannot be suppressed simultaneously. Moreover, [19] developed the independent control of RSC and GSC for DFIG system, in which GSC is responsible for the sinusoidal GSC current, while RSC is responsible for the different targets, i.e., sinusoidal stator current, constant power, or steady torque.

Nevertheless, the time delay was ignored in [18,19], which may influence the performance of harmonics suppression for the large scale DFIG system with low switching frequency in practice. More importantly, the control of RSC and GSC in [18,19] is independent, which means when constant power or steady torque is assigned as the control target of RSC, stator current will be distorted so that the power quality of the total DFIG system cannot be guaranteed [19,20]. Thus, there is a trade-off between power quality and generator performance in the DFIG system with independent control.

For guaranteeing that DFIG have the capability of covering generator performance and power quality simultaneously, this paper develops a collaborative control for DFIG system based on the improved harmonics suppression controller. The proposed control strategy is described in detail as follows:

(1)　The improved harmonics suppression controller is proposed with the consideration of time delay. The time delay of the digital control system can lag the phase of controller [10,11], thereby degrading the control performance [13], especially for the DFIG system with low switching frequency. So, it is essential to compensate the time delay to improve the practicability of harmonics suppression controller, while [18,19] did not cover the issue of time delay.

(2)　For improving performance of generator on the basis of maintaining power quality of DFIG system, a collaborative control strategy is developed in this paper. In the collaborative control, RSC is responsible for one of three different control targets, i.e., sinusoidal stator current, constant power, and constant generator torque, while GSC is responsible for compensating the current distortion of the generator to guarantee the sinusoidal total current of DFIG system.

This paper will be presented as follows, in Section 2, the mathematical model of DFIG system under distorted grid is illustrated briefly. The collaborative control strategy and the design procedures of the harmonics suppression controller are described in Section 3. Section 4 analyzes the control performance and stability of the DFIG system with the proposed collaborative control strategy. Experimental results derived using the 1 kW DFIG system are shown in Section 5. Section 6 gives the conclusions of this paper.

## 2. Model of DFIG System under Harmonically Distorted Grid

As illustrated in Figure 1, the two converters of DFIG system are connected via a DC bus. RSC is responsible for achieving the accurate power control to realize maximum power point tracking, while GSC controls the DC voltage to guarantee the operation of RSC. The model of RSC and GSC should be deduced firstly, since they form the bases of operation analysis for the DFIG under grid harmonics.

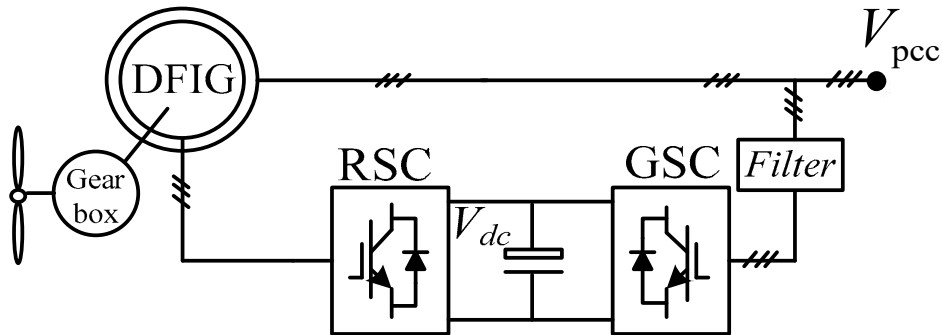

**Figure 1.** Structure of the doubly fed induction generator (DFIG ) system.

*2.1. Model of Induction Generator and RSC*

According to [6–8], the equivalent circuit of generator can be shown in Figure 2, the voltage equation and flux linkage equation of the induction generator can be described as follows:

$$\begin{cases} \boldsymbol{U_{sdq}} = \frac{d\psi_{sdq}}{dt} + j\omega_1\boldsymbol{\psi_{sdq}} + R_s\boldsymbol{I_{sdq}} \\ \boldsymbol{U_{rdq}} = \frac{d\psi_{rdq}}{dt} + j\omega_{slip}\boldsymbol{\psi_{rdq}} + R_r\boldsymbol{I_{rdq}} \end{cases} \tag{1}$$

$$\begin{cases} \boldsymbol{\psi_{sdq}} = L_m\boldsymbol{I_{rdq}} + L_s\boldsymbol{I_{sdq}} \\ \boldsymbol{\psi_{rdq}} = L_m\boldsymbol{I_{sdq}} + L_r\boldsymbol{I_{rdq}} \end{cases} \tag{2}$$

where, $\omega_1$, $\omega_r$, $\omega_{slip}$ are synchronous speed, rotor speed and slip; $L_s$ and $L_r$ represent the stator inductance and rotor inductance; $R_s$ and $R_r$ are the equivalent resistance; $L_m$ is magnetic inductance; $\boldsymbol{U_{sdq}}$, $\boldsymbol{U_{rdq}}$ and $\boldsymbol{I_{sdq}}$, $\boldsymbol{I_{rdq}}$ respectively represent voltage and current of stator side and rotor side, while $\boldsymbol{\psi_{sdq}}$ and $\boldsymbol{\psi_{rdq}}$ represent the flux of stator winding and rotor winding.

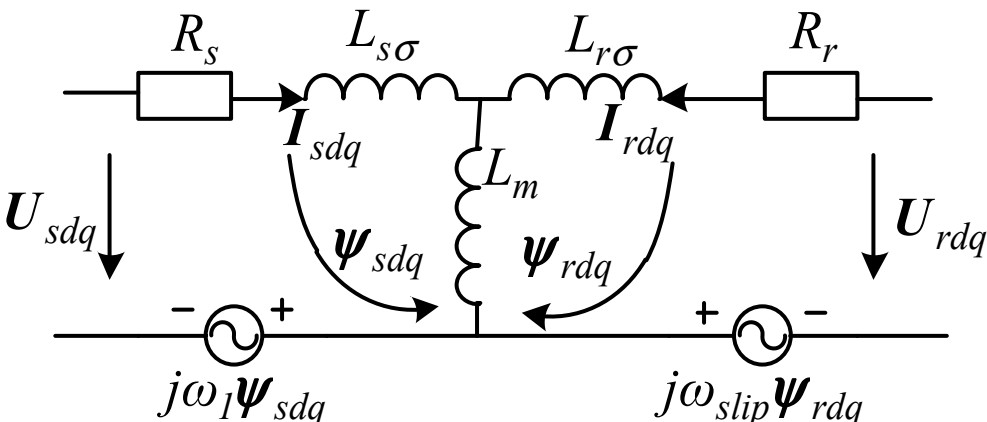

**Figure 2.** Equivalent circuit of generator.

By ignoring the resistance of generator [6–8], the voltage of RSC can be expressed as:

$$\boldsymbol{U_{rdq}} = j\omega_{slip}\sigma L_r\boldsymbol{I_{rdq}} + \sigma L_r\frac{d\boldsymbol{I_{rdq}}}{dt} + \frac{L_m}{L_s}(\boldsymbol{U_{sdq}} - j\omega_r\boldsymbol{\psi_{sdq}}) \tag{3}$$

where $\sigma$ is defined as $1-L_m{}^2/(L_sL_r)$.

Thereby, the active power $P_s$, the reactive power $Q_s$ and the electromagnetic torque $T_e$ can be obtained as,

$$\begin{cases} P_s = 1.5\text{Re}(\boldsymbol{U}_{sdq}\hat{\boldsymbol{I}}_{sdq}) \\ Q_s = 1.5\text{Im}(\boldsymbol{U}_{sdq}\hat{\boldsymbol{I}}_{sdq}) \\ T_e = -1.5n_p\text{Im}(\boldsymbol{\psi}_{sdq}\hat{\boldsymbol{I}}_{sdq}) \end{cases} \tag{4}$$

where, $n_p$ represents the pole-pairs, the superscript ^ represents the conjugate vector.

As in Equations (3) and (4), when $\boldsymbol{U}_{sdq}$ contains harmonics, it can distort the rotor current and the stator current, thereby the rillples will exist in $P_s$, $Q_s$ and $T_e$. Even when current distortions are eliminated thoroughly, the ripples still exist in $P_s$, $Q_s$ and $T_e$, which indicates that the current distortions, power ripples and torque ripples cannot be eliminated simultaneously. Hence, the harmonics suppression controller with alternative control targets is significant for the DFIG system.

*2.2. Model of GSC*

The voltage of GSC can be written as [21,22]:

$$\boldsymbol{U}_{cdq} = -(R_g + j\omega_1 L_g)\boldsymbol{I}_{gdq} - L_g\frac{d\boldsymbol{I}_{gdq}}{dt} + \boldsymbol{U}_{gdq} \tag{5}$$

where, $R_g$ and $L_g$ are the equivalent resistance and equivalent inductance of the filter, $\boldsymbol{I}_{gdq}$ represents the current of GSC side, while $\boldsymbol{U}_{cdq}$ and $\boldsymbol{U}_{gdq}$ are the output voltage of converter and grid voltage, respectively.

When grid voltage contains harmonics, the harmonics exist in output current of GSC as well. Therefore, when the inter-harmonics exist in the grid voltage, the inter-harmonics component with the corresponding frequency will be generated in the output of current DFIG system. The distorted current is unsatisfactory for the requirements of grid connection. Meanwhile the ripples in power and generator torque can worsen the operation performance and reduce the lifetime of the generator.

## 3. The Coordinated Control Strategy and Design Procedure of The Proposed Controller

The traditional PI controller cannot have enough bandwidth to cover the frequency range of grid harmonics, moreover the existing harmonics suppressors are only valid for the integer harmonics. To improve performance of DFIG system under grid harmonics considering inter-harmonics, this paper introduces a harmonics controller to extend the bandwidth of harmonics control.

The proposed controller contains a high-pass filter, a lead element and a time delay compensation, in which the high-pass filter is used to isolate fundamental control and harmonics control, while the lead element is employed to counteract the lag characteristic of the R-L circuit in the range of harmonic frequency. Also, the controlled variables of the controller can be alternatively different including output current, output power and generator torque.

Thereby, a coordinated control strategy is developed based on the harmonics suppression controller, in which the control targets for RSC can be flexibly selected at sinusoidal stator current, smooth power, or constant generator torque. When the control targets of RSC are assigned as smooth power or constant torque, the output current of the DFIG stator cannot be sinusoidal, thus GSC is responsible for guaranteeing the sinusoidal total current for DFIG system.

*3.1. Schematic Diagram of the Proposed Control Strategy*

Figure 3 illustrates the scheme of the proposed control strategy for. The two control loops in the proposed control strategy can be described as follows.

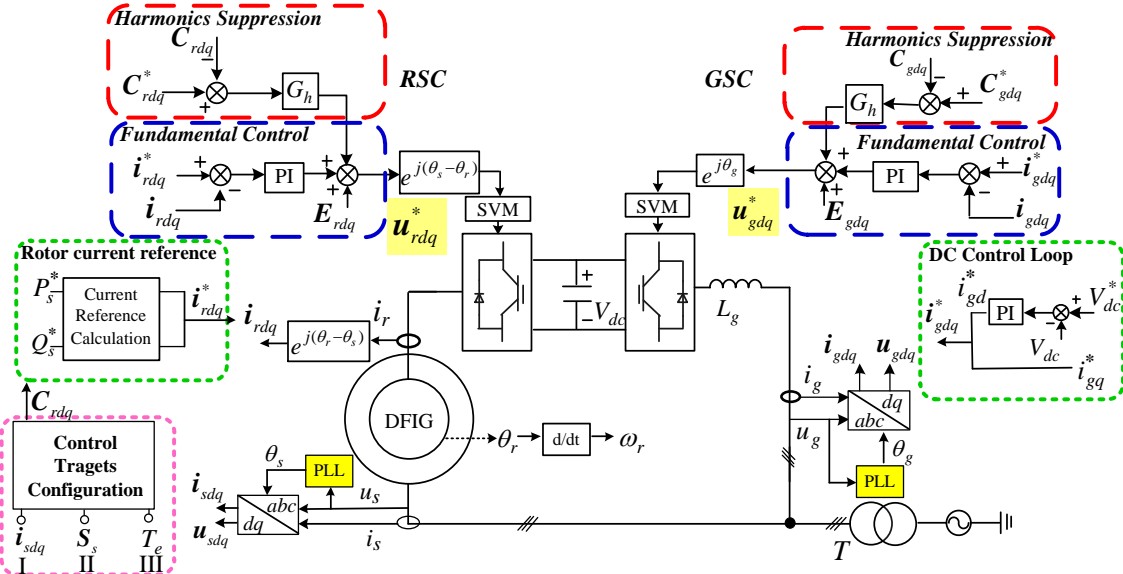

**Figure 3.** The coordinated control strategy with the harmonics suppression controller for DFIG system.

### 3.1.1. Fundamental Controller

The fundamental controller is implemented based on the traditional vector control, in which the phase locked loop in the synchronous reference frame (SRF-PLL) [23] are employed to obtain the frequency and electrical angle of grid voltage. For RSC and GSC, the current loops are regulated by the PI controller to achieve the fundamental current tracking, which can be referred in [7–9]. The current reference of RSC is generated by the power reference as shown in (6), while the current reference of GSC is generated by an outer DC control loop to maintain the stable DC bus voltage [21,22].

$$
\begin{cases}
i_{rd}^* = -\dfrac{2L_s P_s^*}{3L_m u_{sd}} \\[2mm]
i_{rq}^* = -\dfrac{2L_s Q_s^*}{3L_m u_{sd}} - \dfrac{u_{sd}}{\omega_g L_s}
\end{cases}
\tag{6}
$$

### 3.1.2. Harmonics Suppression Controller

Besides the fundamental control loop, the additional harmonics control loops are introduced into RSC and GSC to achieve the harmonics suppression for DFIG system. The control target of RSC can be different in a sinusoidal stator current, constant output power, or steady generator torque, and to achieve different performance requirements, as shown in the control targets configuration of Figure 3. With the different control targets, $C_{rdq}$ of RSC can be obtained as,

Target I: Sinusoidal stator current

$$
C_{rdq} = i_{sdq} = i_{sd} + i_{sq}j
\tag{7}
$$

Target II: Constant output power

$$
C_{rdq} = S_s = P_s + Q_s j
\tag{8}
$$

Target III: steady generator torque

$$
C_{rdq} = T_e
\tag{9}
$$

The control target of GSC is the sinusoidal total current, so $C_{gdq}$ is set as total current.

$$
C_{gdq} = i_{tdq} = i_{td} + i_{tq}j
\tag{10}
$$

Accordingly, the reference voltage of RSC and GSC can be expressed as:

$$
\begin{cases}
\boldsymbol{u}^*_{rdq} &= \boldsymbol{u}^*_{rPI} + \boldsymbol{u}^*_{rh} + \boldsymbol{E}_{rdq} \\
&= H_{ri}(\boldsymbol{i}^*_{rdq} - \boldsymbol{i}_{rdq}) + G_h(T^*_{rdq} - T_{rdq}) + E_{rdq} \\
\boldsymbol{u}^*_{gdq} &= \boldsymbol{u}^*_{gPI} + \boldsymbol{u}^*_{gh} + \boldsymbol{E}_{gdq} \\
&= H_{gi}(\boldsymbol{i}^*_{gdq} - \boldsymbol{i}_{gdq}) + G_h(T^*_{gdq} - T_{gdq}) + E_{gdq}
\end{cases}
\tag{11}
$$

where, $\boldsymbol{C}^*gdq$ and $\boldsymbol{C}^*rdq$ are set as zero, $G_h$ is the proposed harmonics suppression controller, the design procedure of which will be described in next section. $H_{ri}$ and $H_{gi}$ are the current controller of RSC and GSC. $\boldsymbol{E}_{rdq}$ and $\boldsymbol{E}_{gdq}$ can be expressed as Equation (12), which shows the decoupling components of RSC and GSC.

$$
\begin{cases}
\boldsymbol{E}_{rdq} = j\omega_{slip}\sigma L_r\boldsymbol{I}_{rdq} + \frac{L_m}{L_s}(\boldsymbol{U}_{sdq} - j\omega_r\boldsymbol{\psi}_{sdq}) \\
\boldsymbol{E}_{gdq} = -j\omega_1 L_g\boldsymbol{I}_{gdq} + \boldsymbol{U}_{gdq}
\end{cases}
\tag{12}
$$

### 3.2. Design Procedure of Harmonics Suppression Controller

As mentioned above, the proposed harmonics suppression controller contains three parts, which are a high-pass filter, a modified differential element and a compensation of time delay. Thereby, the expression $G_h(s)$ can be written as,

$$
G_h(s) = G_{filter}(s) \times G_{lead}(s) \times G_{com\_delay}(s)
\tag{13}
$$

#### 3.2.1. High-Pass Filter

The existing controller for harmonics suppression is based on resonator basically, however the resonator is only valid at the fixed frequency actually, so the existing methods are invalid for the inter-harmonics.

The controller should be capable to suppress the integer and inter-harmonic components in a wide frequency range and will not influence the fundamental control. The Chebyshev high-pass filter has the characteristic of narrow steep zone, which can guarantee the harmonics suppression loop barely affects fundamental control [24]. Considering practicability, the order of the filter should not be overly high, thus a second order filter is employed in this paper. Since the considered harmonics frequency are higher than 250 Hz, the cut-off frequency of the filter should be lower than 250 Hz, $G_{filter}(s)$ can be obtained as (14), where $\omega_n = 200 \, \pi$rad/s.

$$
G_{filter}(s) = \frac{0.989s^2}{s^2 + 0.716\omega_n s + (0.302\omega_n)^2}
\tag{14}
$$

#### 3.2.2. Modified Differential Element

Since the equivalent model of DFIG can be regarded as a series R-L circuit, which means the controlled object is a first order inertial element. In order to counteract the lag characteristic of the R-L circuit, a differential (lead) element should be introduced. In view of the conventional differential element is non-causal, usually a low-pass filter of which cut-off frequency is much higher than the specific frequency range will be attached to the differential element, so that the modified differential element can be implemented in practice. Considering the harmonics frequency range is about 250 Hz to 1000 Hz, the cut-off frequency $\omega_c$ of the lowpass filter is designed as 3000 $\pi$rad/s. The expression of $G_{lead}(s)$ can be written as:

$$
G_{lead}(s) = \frac{s}{s + \omega_c}
\tag{15}
$$

### 3.2.3. Compensation of System Delay

The controller should be effective in a wide frequency range, which means the time delay of the control system cannot be neglected. The time delay usually considered as 1.5 times sampling period $(T_s)$, thus the ideal compensation can be expressed as $e^{1.5sT_s}$, however it is non-causal which means it cannot be implemented in practice. On the occasion of $\omega_1$ is much larger than $\omega_2$, as shown in (16), the lead-lag compensator can be used to achieve the compensation effect in the specific frequency range, thus the expression of delay compensation can be written as (16). The compensation angle $\theta_{com\_delay}$ can be expressed as Equation (17) approximately, the $\omega_2$ can be set as $1/(1.5T_s)$, while $\omega_1$ should be much larger than $\omega_2$ and can be selected as 100,000 $\pi$rad/s in this paper.

$$G_{com\_delay}(s) = e^{1.5sT_s} \approx \frac{1 + s/\omega_2}{1 + s/\omega_1} \tag{16}$$

$$\theta_{com\_delay} \approx \omega\left(\frac{1}{\omega_2} - \frac{1}{\omega_1}\right) \approx \frac{\omega}{\omega_2} \tag{17}$$

## 4. The Analyses of Control Performance for The DFIG System with The Coordinated Control

### 4.1. Analysis for the System Delay Compensation

In order to analyze the effect of the system delay compensation, $G_{h\_design}$ is defined to represent the expected characteristic of the harmonics suppression controller, $G_{h\_delay}$ is defined to represent the characteristic of the controller without delay compensation, $G_{h\_com}$ is defined to represent the characteristics of the controller with delay compensation, the expressions of which can be written as:

$$G_{h\_design}(s) = G_{filter}(s) \times G_{lead}(s) \tag{18}$$

$$G_{h\_delay}(s) = G_{filter}(s) \times G_{lead}(s) \times e^{-1.5sT_s} \tag{19}$$

$$G_{h\_com}(s) = G_{filter}(s) \times G_{lead}(s) \times G_{com\_delay}(s)e^{-1.5sT_s} \tag{20}$$

It was indicated that the harmonics suppression capability becomes stronger when the phase of harmonics controller is closer to 90° [13]. As shown in Figure 4, harmonics suppression capability will be degraded due to reason that the phase lag of the improved controller is gradually increasing with the frequency rising. The introduction of the delay compensation can counteract the phase lag which caused by a system delay in the frequency range from 100 Hz to 1000 Hz effectively. Thus, the usage of the time delay component can guarantee the performance of the harmonics suppression controller will not be worsen by the system delay.

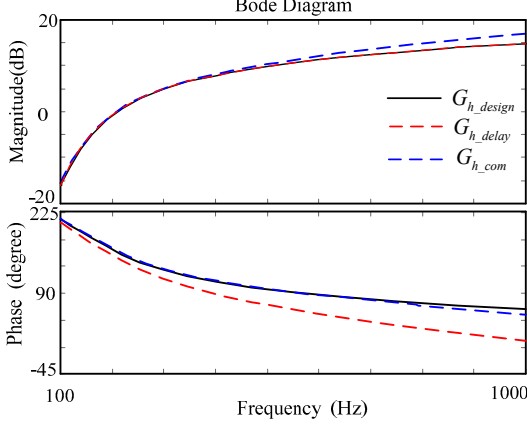

**Figure 4.** Bode diagram of the harmonics suppression controller.

### 4.2. Performance Analysis for Harmonics Suppression

As indicated by Equations (3)–(5), the control block of RSC and GSC can be presented as Figure 5a,b respectively, the transfer function from control object to grid voltage under different control targets can be expressed as *F*. Thus, *F* can be used to analyze the anti-disturbance capability of different control object to the grid harmonics. The smaller the magnitude of *F*, the stronger anti-disturbance capability can be obtained.

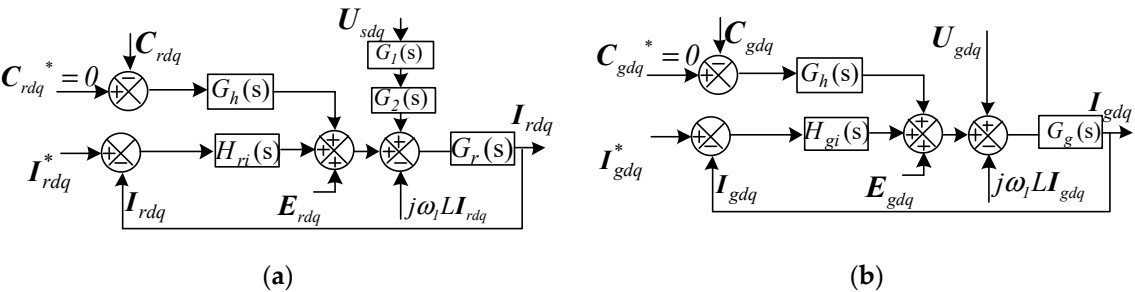

(a)                          (b)

**Figure 5.** Control Block of DFIG system with the harmonics suppression. (**a**) Control Block of RSC (**b**) Control Block of GSC.

As can be seen in Equations (21)–(26), different control targets are distinguished by the superscript *I*, *II*, *III*. Further, subscripts *rsc* and *gsc* represent the variables in RSC or GSC, in which $G_1(s) = 1/(s + j\omega_1)$, $G_2(s) = L_m(s + j\omega_{slip})/L_s$, $G_3(s) = 3L_mu_{sd}/2L_s$, $G_4(s) = 3L_m\sigma u_{sd}/2L_s$, $G_5(s) = -1.5n_p\psi_{sq}$, $G_6(s) = \psi_{sq}/L_s$, $G_r(s) = 1/(R_r + \sigma L_r s)$, $G_g(s) = 1/(R_g + L_g s)$.

$$F_{rsc}^{I} = \frac{i_{sdq}}{u_{sdq}} = \frac{G_1(s)[1 + H_{ri}(s)G_r(s)] + G_1(s)G_2(s)G_r(s)L_m}{L_s + [L_mH_{ri}(s) + L_sG_h(s)]G_r(s)} \tag{21}$$

$$F_{gsc}^{I} = \frac{i_{tdq}}{u_{sdq}} = \frac{i_{sdq} + i_{gdq}}{u_{sdq}} = \frac{G_g(s) + [1 + H_{gi}(s)G_g(s)]F_{rsc}^{I}(s)}{1 + [H_{gi}(s) + G_h(s)]G_g(s)} \tag{22}$$

$$F_{rsc}^{II} = \frac{S_{sdq}}{u_{sdq}} = \frac{G_1(s)\{G_2(s)G_r(s)G_3(s) - G_4(s)[1 + H_{ri}(s)G_r(s)]\}}{1 + [H_{ri}(s) + G_3(s)G_h(s)]G_r(s)} \tag{23}$$

$$F_{gsc}^{II} = \frac{i_{tdq}}{u_{sdq}} = \frac{G_g(s) + [1 + H_{gi}(s)G_g(s)]L_mF_{rsc}^{II}(s)/L_sG_3(s)}{1 + [H_{gi}(s) + G_h(s)]G_g(s)} \tag{24}$$

$$F_{rsc}^{III} = \frac{T_e}{u_{sdq}} = \frac{-G_1(s)G_2(s)G_r(s)G_5(s)}{1 + [H_{ri}(s) + G_5(s)G_h(s)]G_r(s)} \tag{25}$$

$$F_{gsc}^{III} = \frac{i_{tdq}}{u_{sdq}} = \frac{G_g(s) + [1 + H_{gi}(s)G_g(s)][F_{rsc}^{III}(s)/G_5(s) + G_6(s)]}{1 + [H_{gi}(s) + G_h(s)]G_g(s)} \tag{26}$$

When the control target of RSC is activated as Sinusoidal stator current, $F_{rsc}^{I}$ and $F_{gsc}^{I}$ represent anti-disturbance capability of the stator current and total current to grid harmonics respectively, of which bode diagram can be seen in Figure 6a. After enabling the proposed control, at 300 Hz and 600 Hz, where are the frequencies of traditional 5th and 7th integer harmonics, the magnitude of $F_{rsc}^{I}$ can be decreased from −7.8 dB and −13.2 dB to −31 dB and −35.1 dB, while the magnitude of $F_{gsc}^{I}$ can be decreased from −11.3 dB and −14.7 dB to −32.5 dB and −36.1 dB. In fact, besides the frequency of integer harmonics, the magnitude of $F_{rsc}^{I}$ and $F_{gsc}^{I}$ are significantly decreased in the whole range from 100 Hz to 1000 Hz, which indicates that the integer harmonics and inter-harmonics in stator current and total current of DFIG system both can be effectively suppressed.

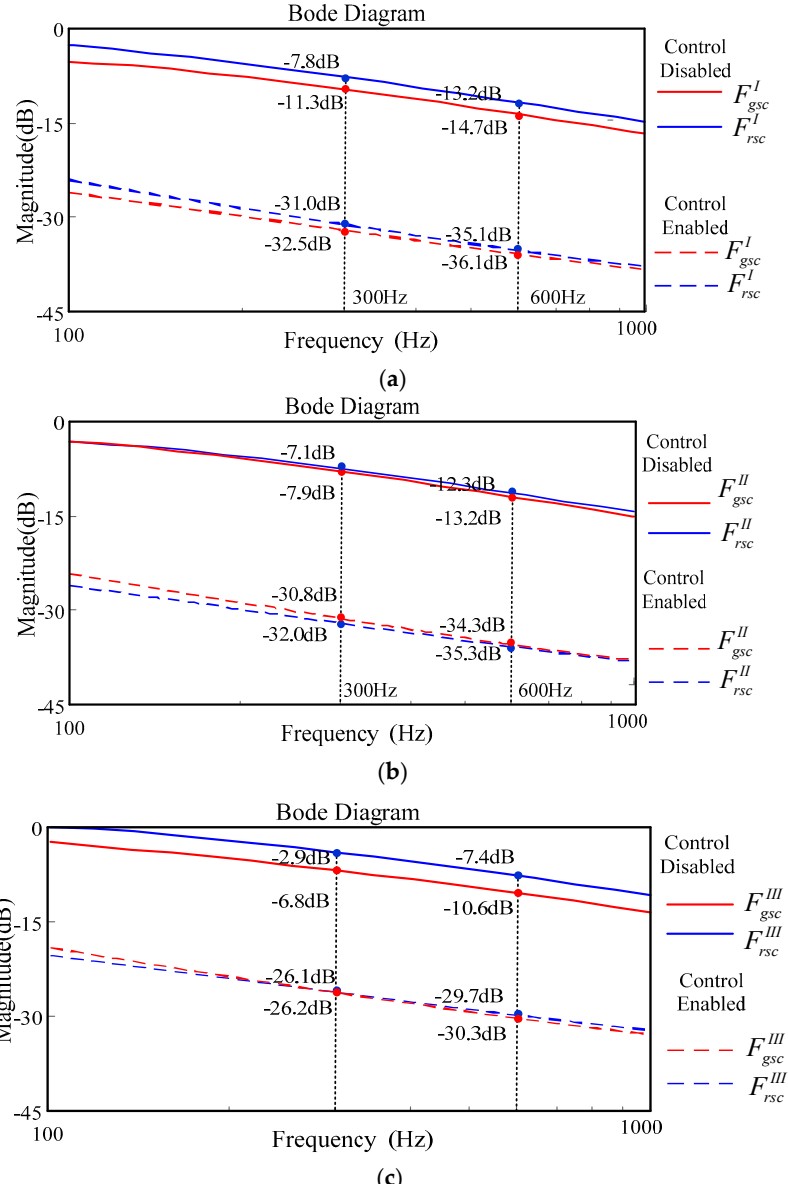

**Figure 6.** Bode diagram of *F* when different control targets are applied. (**a**) Control target of RSC is active as target I, sinusoidal stator current; (**b**) Control target of RSC is active as target II, constant power; (**c**) Control target of RSC is active as target III, steady generator torque.

Likewise, when the control target of RSC is activated as constant power, $F_{rsc}^{II}$ and $F_{gsc}^{II}$ represent anti-disturbance capability of the generator power and total current to grid harmonics respectively. As shown in Figure 6b, the magnitude of $F_{rsc}^{II}$ can be decreased from −7.1 dB and −12.2 dB to −32 dB and −35.3 dB, respectively, while the magnitude of $F_{gsc}^{II}$ can be decreased from −7.9 dB and −13.2 dB to −30.8 dB and −34.3 dB at 300 Hz and 600 Hz, respectively. Likewise, the magnitude of $F_{rsc}^{II}$ and $F_{gsc}^{II}$ can be decreased significantly at the frequency besides integer harmonics as well in the range from 100 Hz to 1000 Hz.

When the control target of steady torque is selected at RSC, as shown in Figure 6c. The magnitude of $F_{rsc}^{III}$ can be decreased from −2.9 dB and −6.8 dB to −26.1 dB and −26.2 dB at 300 Hz and 600 Hz, while the magnitude of $F_{gsc}^{III}$ can be decreased from −7.4 dB and −10.6 dB to −29.7 dB and −30.3 dB, respectively. Likewise, the decrements of the magnitude of $F_{rsc}^{III}$ and $F_{gsc}^{III}$ are significant in the frequency range from 100 Hz to 1000 Hz, which indicates that torque ripples and total current harmonics can be suppressed significantly.

### 4.3. Performance Analyses for Fundamental Control

With the block diagram of RSC and GSC, the transfer functions from rotor current to current reference and the transfer function from GSC current to current reference are expressed as *H*, which can be employed to reveal the fundamental control performance of RSC and GSC. Equations (27)–(30) give the expressions of *H*.

$$H_{rsc}^{I} = \frac{\boldsymbol{i}_{rdq}}{\boldsymbol{i}_{rdq}^{*}} = \frac{H_{ri}(s)G_r(s)}{1 + [H_{ri}(s) + G_h(s)]G_r(s)} \tag{27}$$

$$H_{rsc}^{II} = \frac{\boldsymbol{i}_{rdq}}{\boldsymbol{i}_{rdq}^{*}} = \frac{H_{ri}(s)G_r(s)}{1 + [H_{ri}(s) + G_3(s)G_h(s)]G_r(s)} \tag{28}$$

$$H_{rsc}^{III} = \frac{\boldsymbol{i}_{rdq}}{\boldsymbol{i}_{rdq}^{*}} = \frac{H_{ri}(s)G_r(s)}{1 + [H_{ri}(s) + G_5(s)G_h(s)]G_r(s)} \tag{29}$$

$$H_{gsc}^{I} = H_{gsc}^{II} = H_{gsc}^{III} = \frac{\boldsymbol{i}_{gdq}}{\boldsymbol{i}_{gdq}^{*}} = \frac{G_g(s)H_{gi}(s)}{1 + [H_{gi}(s) + G_h(s)]G_g(s)} \tag{30}$$

As illustrated in Figure 7, the introduction of the harmonics suppression controller will not affect accuracy of the fundamental control of RSC and GSC, since the characteristics of unit gain and zero phase of $H_{rsc}$ and $H_{gsc}$ can be maintained when the different control targets are selected. Meanwhile, it should be noted that the cut-off frequency of $H_{rsc}$ will be decreased to 49 Hz, 44 Hz, and 41 Hz respectively, from 84 Hz when the different control targets are applied, which indicates that the dynamic response of RSC will be slowed down to different degrees after employing the harmonics suppression with different control targets. As to GSC, the dynamic response will be slowed down as well, since the cut-off frequency of $H_{gsc}$ is reduced to 116 Hz from 316 Hz.

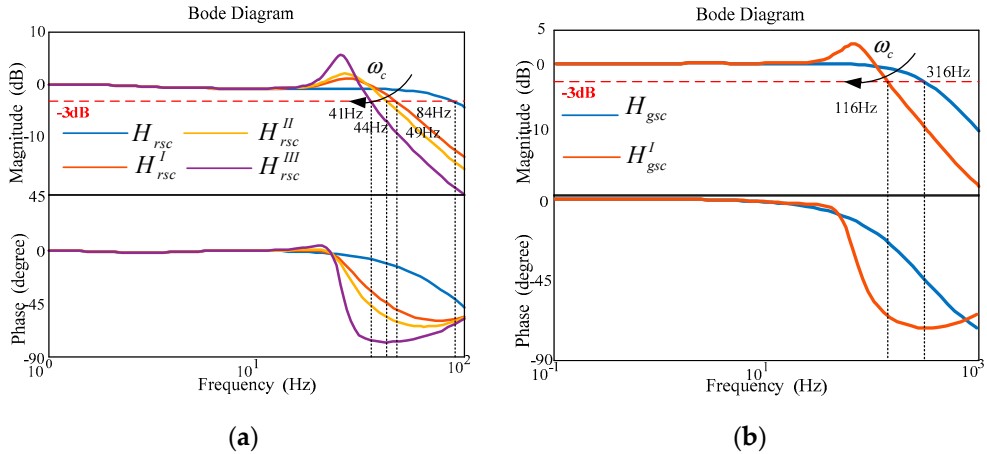

(a)  (b)

**Figure 7.** Bode diagram of *H* when different control targets are applied. (**a**) Fundamental control of RSC; (**b**) Fundamental control of GSC.

## 5. Experimental Results for The Proposed Control Strategy

For the sake of verifying the feasibility and practicability of the proposed collaborative strategy, a 1 kW DFIG system was built to achieve the experimental verifications, the parameters of the DFIG are given in Table 1, while the grid side inductance and resistance are 2 mH and 0.01 Ω, respectively.

### 5.1. Verification for Integer Harmonics Suppression

Figure 8 illustrates the operation of the DFIG system under integer harmonics grid, of which harmonics components are 2.5%, 2.25%, 1.5% and 1.25% at 250 Hz, 350 Hz, 550 Hz and 650 Hz respectively. On this occasion, the power quality and operation performance of DFIG system is

deteriorated severely if the proposed control method is disabled, as shown in Figure 8a, the THD of total current is 5.07%, in which the components at 250 Hz, 350 Hz, 550 Hz and 650 Hz are 3.26%, 2.84%, 1.71% and 1.36% respectively. The THD of the stator current is 4.93%, and the THD of components at 250 Hz, 350 Hz, 550 Hz and 650 Hz are 3.11%, 2.74%, 1.60% and 1.35%, respectively. Meanwhile, the contents of ripples in the active power, reactive power, and generator torque are 3.98%, 5.87%, and 4.71%.

**Table 1.** Parameters of DFIG system.

| DFIG Parameters | | | | |
|---|---|---|---|---|
| $P_n$/kW | $U_n$/kV | $f_n$/Hz | $V_{dc}$/kV | $L_m$/mH |
| 1 | 0.11 | 50 | 0.25 | 91 |
| $R_s$/ohm | $L_{\sigma1}$/mH | $R_r$/ohm | $L_{\sigma2}$/mH | |
| 1.01 | 3 | 0.9 | 3.01 | |

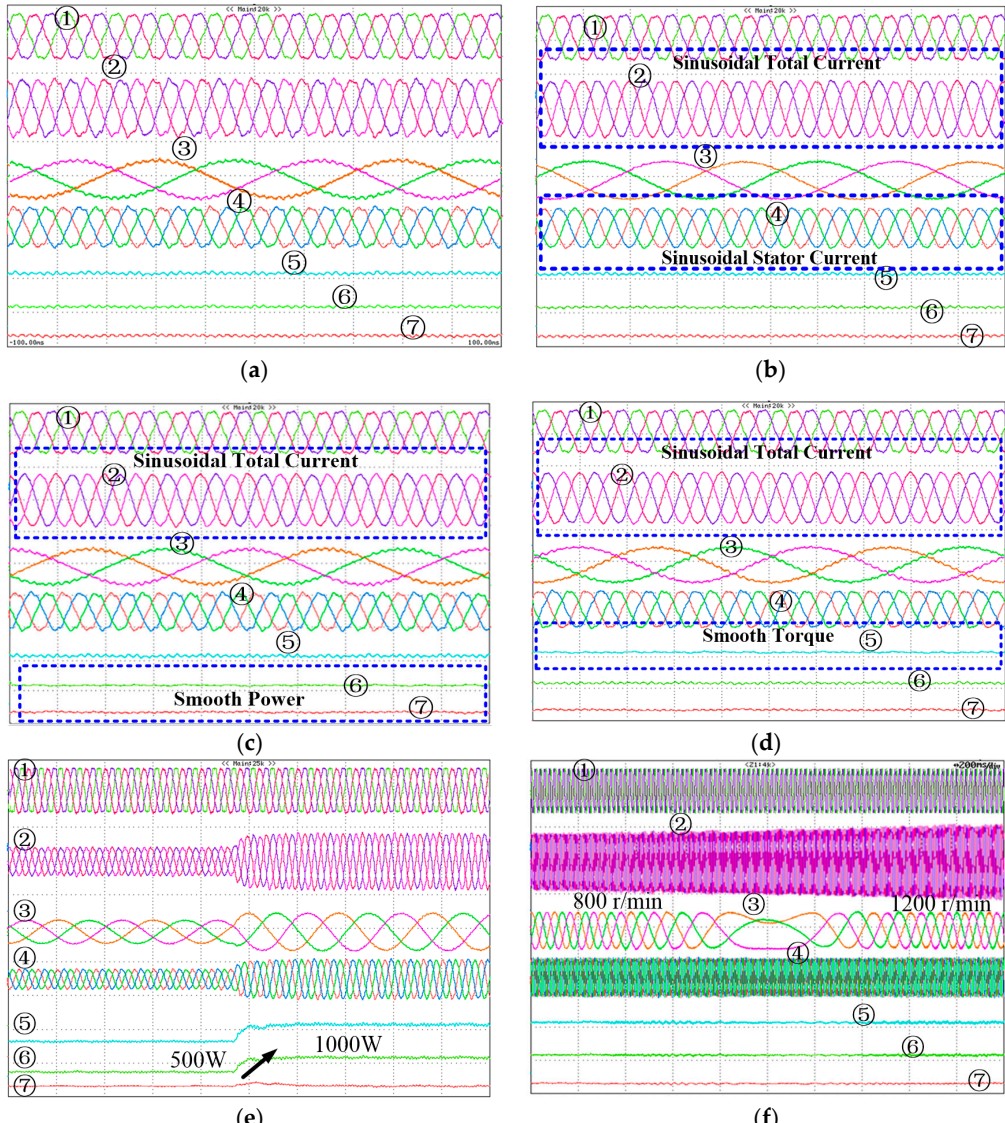

**Figure 8.** Operation of DFIG system under integer grid harmonics [①Stator Voltage $U_{sabc}$ (250 V/div), ② Total Current $I_{tabc}$ (5 A/div), ③Rotor Current $I_{rabc}$ (5 A/div), ④ Stator Current $I_{sabc}$ (10 A/div), ⑤ Generator Torque $T_e$ (12 N.m/div), ⑥Active Power $P_s$ (1.5 kW/div), ⑦ Reactive Power $Q_s$ (1.5 kVar/div)]. (**a**) Without the proposed harmonics suppression. (**b**) Enabled with Target I. (**c**) Enabled with Target II. (**d**) Enabled with Target III. (**e**) Power regulation of DFIG system. (**f**) Speed regulation of DFIG system.

When the coordinated control strategy is activated, GSC guarantees the total current is sinusoidal, while the target of RSC can be selected flexibility. Figure 8b–d shows the results of the DFIG system when different control targets of RSC are applied. When the target of RSC is selected as the sinusoidal stator current, the components integer harmonics in stator current are reduced to 0.91% 0.74%, 0.51%, and 0.43%, while the THD of total current can be suppressed to 1.44% as well. When the target of RSC is selected as the smooth power, the active power ripple and reactive power ripple can be reduced to 1.03% and 1.58%, while the THD of total current can be maintained as 1.29%. When the target of RSC is selected as the smooth torque, the torqued ripples can be decreased to 1.08%, while the THD of total current can be maintained as 1.38%. Table 2 gives the experimental results of the proposed control strategy under integer harmonics grid. It can be concluded that the DFIG system with the proposed controller can operate well with different control targets, while the power quality of the total system can be guaranteed to satisfy the grid connection requirement.

**Table 2.** Control performance of the proposed control strategy under grid integer harmonics.

|  | *None* | **Target I** | **Target II** | **Target III** |
|---|---|---|---|---|
| *THD of $I_{tabc}$* | 5.07% | 1.44% | 1.29% | 1.38% |
| *THD of $I_{sabc}$* | 4.93% | 1.33% | 4.79% | 3.77% |
| *Ripple of $P_s/Q_s$* | 3.98%/5.87% | 4.04%/5.78% | 1.03%/1.58% | 4.12%/2.05% |
| *Ripple of $T_e$* | 4.71% | 4.54% | 4.99% | 1.08% |

Figure 8e,f gives the power regulation and speed regulation of the DFIG system after equipping the proposed control. The power regulation illustrates that the power step from 500 W to 1 kW can be achieved accurately within 10 ms. The speed regulation illustrates that when the rotor is speeded up from 800 r/min to 1200 r/min (synchronous speed is 1000 r/min), power generation can be maintained well during the regulation.

*5.2. Verification for Inter-Harmonics Suppression*

Figure 9 illustrates performance of the proposed control method under grid inter-harmonics, the components of the inter-harmonics in grid voltage at 260 Hz, 364 Hz, 572 Hz, and 676 Hz are 2.5%, 2.2%, 1.25%, and 1.3%, respectively. Similar to Figure 8a, Figure 9a illustrates that the operation of DFIG system is unsatisfactory under gird inter-harmonics, the THD of total current is 4.72%, in which the components at 260 Hz, 364 Hz, 572 Hz and 676 Hz are 2.86%, 2.42%, 1.64%, and 1.12%. Meanwhile ripples exist in active power, reactive power, and generator torque, of which components are 4.05%, 5.75%, and 4.94%, respectively.

As can be seen in Figure 9b–d, the proposed control strategy is effective for DFIG system under grid inter-harmonics. When the control target I is selected, the harmonics of stator current can be suppressed significantly from 4.70% to 1.36%, in which the components at 260 Hz, 364 Hz, 572 Hz and 676 Hz are reduced from 2.86%, 2.42%, 1.64%, and 1.12% to 1.03%, 0.71%, 0.43%, and 0.40%, respectively. The THD of the total current is suppressed to 1.40%. When the control target II is selected, the ripples in active power and reactive power can be suppressed to 0.99% and 1.55%, while the THD of total current can be maintained as 1.32%. As to the control target III, the ripple of generator torque is reduced to 1.10%, while the THD of total current can be suppressed to 1.34%. Table 3 gives the experimental results analysis under grid inter-harmonics, which indicates that the proposed coordinated control has the capability of improving the performance and the flexibility of harmonics suppression simultaneously. Likewise, Figure 9e,f illustrates that the power regulation and speed regulation for DFIG system can be achieved, which indicates that the coordinated control can improve the operation of DFIG system under grid inter-harmonics as well as integer harmonic.

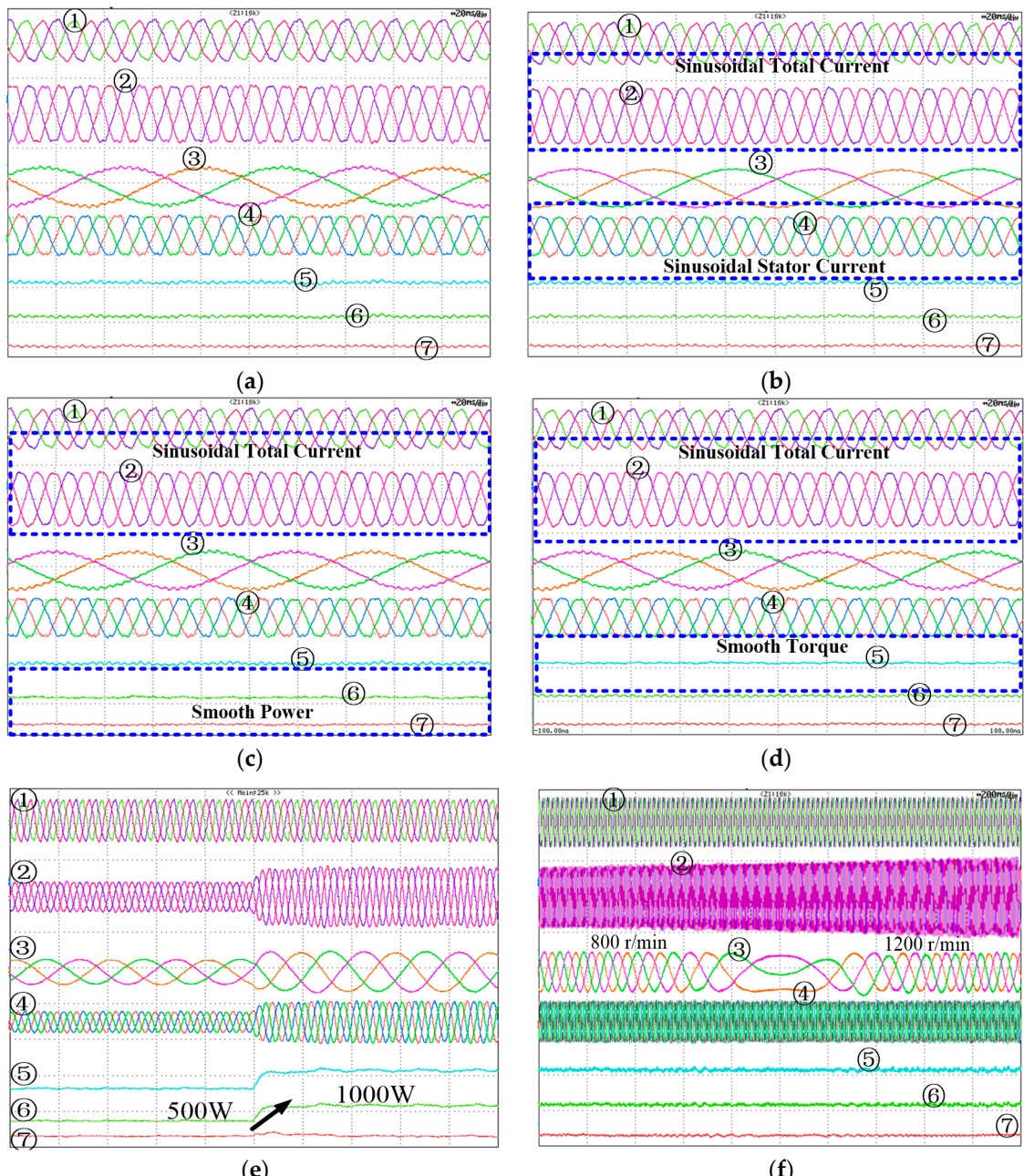

**Figure 9.** Operation of the DFIG system under grid inter-harmonics [① Stator Voltage $U_{sabc}$ (250 V/div),
② Total Current $I_{rabc}$ (5 A/div), ③ Rotor Current $I_{rabc}$ (5 A/div), ④ Stator Current $I_{sabc}$ (10 A/div), ⑤
Generator Torque $T_e$ (12 N.m/div), ⑥Active Power $P_s$ (1.5 kW/div), ⑦ Reactive Power $Q_s$ (1.5 kVar/div)].
(**a**) Without the proposed harmonics suppression. (**b**) Enabled with Target I. (**c**) Enabled with Target II.
(**d**) Enabled with Target III. (**e**) Power regulation of DFIG system. (**f**) Speed regulation of DFIG system.

**Table 3.** Control Performance of the proposed control strategy under grid inter-harmonics.

|  | None | Target I | Target II | Target III |
|---|---|---|---|---|
| THD of $I_{tabc}$ | 4.72% | 1.40% | 1.32% | 1.34% |
| THD of $I_{sabc}$ | 4.70% | 1.36% | 4.72% | 4.41% |
| Ripple of $P_s/Q_s$ | 4.05%/5.75% | 3.94%/5.26% | 0.99%/1.55% | 3.92%/5.64% |
| Ripple of $T_e$ | 4.94% | 4.71% | 4.97% | 1.10% |

Thus, the conclusion which can be summarized by the experimental results and analysis is that the DFIG system with the proposed control strategy can operate with different alternative control targets, while the output power quality of the total system can be guaranteed so that the DFIG system has a flexible operation capability to satisfy the grid connection requirement.

## 6. Conclusions

The contribution of the paper is to introduce a wideband harmonics suppression controller for the DFIG system, which contains a high-pass filter, a modified differential element, and a time delay compensation component, in which the high-pass filter works for extracting the harmonics component, the modified differential element is responsible for counteracting the lag characteristic of the R-L circuit, and the time delay component is employed to improve the feasibility of practical application. Since the proposed controller can effectively work in a wide frequency range from 100 Hz to 1000 Hz, the influences on the operation of the DFIG caused by integer and inter-harmonics voltage can be suppressed effectively.

Furthermore, a coordinated control strategy is proposed for the DFIG system with the proposed coordinated strategy. Different alternative control targets including the sinusoidal output current, constant power, or steady generator torque can be achieved to satisfy different operation requirements for DFIG, while power quality of the total system can be guaranteed when the different control targets are applied.

**Author Contributions:** Conceptualization, B.P. and F.L.; methodology, B.P.; software, H.D.; validation, B.P., H.D. and F.L.; formal analysis, B.P.; investigation, B.P.; resources, H.N.; data curation, H.N.; writing—original draft preparation, B.P.; writing—review and editing, H.N.; visualization, B.P.; supervision, H.N.; project administration, H.N.; funding acquisition, H.N. All authors have read and agreed to the published version of the manuscript.

**Funding:** This work is supported by the National Natural Science Foundation of China under Grant 51622706.

**Conflicts of Interest:** The authors declare no conflict of interest

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
