# Peer review of "Coordinated Control of RSC and GSC for DFIG System under Harmonically Distorted Grid Considering Inter-Harmonics"

_energies, doi:10.3390/en13010028_

Round 1

Reviewer 1 Report

The paper presents a Coordinated Control of RSC and GSC for DFIG System under Harmonically Distorted Grid Considering Inter-harmonics. The authors have clearly stated the purpose of the paper that the time delay is missing in previous research. They also intended to improve the performance of generator on the basis of maintaining power quality of DFIG system by developing collaborative control strategy. They have verified the effectiveness of the proposed control strategy by performing the experiments based on a 1kW DFIG system. Therefore, the paper is recommended for publication after minor corrections in the comments below:

Figure 3 should be moved to, below line 147 after it has been mentioned. The caption for Figure 8 is incomplete, so it is not clear whether it is figure 8 or not. Do the same for Figure 9. Besides, the figures should be place below the paragraphs after they have been mentioned. The conclusion should state clearly the summary of the methods used in the paper. Include the most important summary of results in the conclusion such as the frequency range (as mentioned in the conclusion.

Author Response

The authors appreciate your review. We have carefully revised this paper according to your suggestion, the revised content has been highlighted as red font in the revision paper.

The detailed response can be seen in the attachment named 'Reviewer1'

Reviewer 2 Report

This article describes the proposed control strategy that can better eliminate the harmonics in DFIG systems. Although the theoretical foundations and experimental results are described in detail, the following should be improved:

- Some sentences are too long, so it is difficult to understand their meaning (NPR. Lines 37-41, lines 74-78, lines 227-232, lines 345-349 ...), I suggest dividing it into two or more sentences.

- All pictures, especially 2, 3 and 5, should be enlarged, because in this way the markings on the pictures are too small and difficult to read. Formulas also contain a lot of indexes that are tiny, they need to be scaled up to be minimal in size as tags in the text.

- In addition, the authors do not adhere to the way of marking the chapters and subsections prescribed by the template (not A, B, C, should be 3. then 3.1, or 3.1.1, 3.1.2 ...)

- Figures 3 and 9 should be shown later, after being mentioned in the text.

- The expression (13) is probably a product of the three parts mentioned, but it seems inconspicuous, I suggest that the authors add this to the text and put points in the expression (13).

- The sentence in line 227 should not begin with a number [13]. (maybe The ref. [13] indicates ... or: In the [13] is indicated that ...)

- Abbreviations should be explained on first appearance (THD mentioned for the first time in line 35, full name in line 312)

- The red and green colors added on the text in Tables II and III are too dark, the text cannot be read at all if the print is grayscale.

- Table II should be shown later, after being mentioned in the text.

Author Response

The authors appreciate your review and suggestion, since it can improve the quality of this paper significantly. We have carefully revised this paper according to your suggestion, the revised content has been highlighted as red font in the revision paper.

The detailed response can be seen in the attachment named 'Reviewer2'
